

# Body distribution of impetigo and association with host and pathogen factors

Arvind Yerramilli[1,2], Asha C. Bowen[3,4,5], Adrian J. Marcato[6], Jodie McVernon[6,7,8,9], Jonathan R. Carapetis[3,5], Patricia T. Campbell[6,9] and Steven YC Tong[1,4,6]

[1] Victorian Infectious Diseases Service, The Royal Melbourne Hospital, at the Peter Doherty Institute for Infection and Immunity, Melbourne, Victoria, Australia
[2] Department of Infectious Diseases, Barwon Health, Geelong, Victoria, Australia
[3] Wesfarmers Centre for Vaccines and Infectious Diseases, Telethon Kids Institute, University of Western Australia, Perth, Western Australia, Australia
[4] Menzies School of Health Research, Charles Darwin University, Darwin, Northern Territory, Australia
[5] Department of Infectious Diseases, Perth Children's Hospital, Perth, Western Australia, Australia
[6] Department of Infectious Diseases, The University of Melbourne, at the Peter Doherty Institute for Infection and Immunity, Melbourne, Victoria, Australia
[7] Infection and Immunity, Murdoch Children's Research Institute, Melbourne, Victoria, Australia
[8] Victorian Infectious Diseases Reference Laboratory Epidemiology Unit, The Royal Melbourne Hospital, at the Peter Doherty Institute for Infection and Immunity, Melbourne, Victoria, Australia
[9] Centre for Epidemiology and Biostatistics, Melbourne School of Population and Global Health, Melbourne, Victoria, Australia

Corresponding author
Steven YC Tong,
steven.tong@mh.org.au

## ABSTRACT

**Background:** Impetigo or skin sores are estimated to affect >162 million people worldwide. Detailed descriptions of the anatomical location of skin sores are lacking.
**Methods:** We used prospectively collected data from a randomised control trial of treatments for impetigo in Aboriginal children in Australia. We generated heat-map distributions of skin sores on the human body from 56 predefined anatomical locations and stratified skin sore distribution by sex, age, causative pathogen and co-infection with scabies, tinea and head lice. We compared the distribution of sores between males and females, between sores with only *Streptococcus pyogenes* and sores with only *Staphylococcus aureus*; and across age groups with a Fisher's exact test.
**Results:** There were 663 episodes of impetigo infections among 508 children enrolled in the trial. For all 663 episodes, the lower limbs were the most affected body sites followed by the distal upper limbs, face and scalp. On the anterior surface of the body, the pre-tibial region was the most affected while on the posterior surface, the dorsum of the hands and calves predominated. There was no observable difference between males and females in distribution of sores. Children up to 3 years of age were more likely to have sores on the upper posterior lower limbs and scalp than older age groups, with the distribution of sores differing across age groups ($p = 3 \times 10^{-5}$). Sores from which only *Staphylococcus aureus* was cultured differed in distribution to those with only *Streptococcus pyogenes* cultured ($p = 3 \times 10^{-4}$) and were more commonly found on the upper posterior lower limbs.
**Conclusions:** Skin sores were predominantly found on exposed regions of the lower leg and distal upper limbs. The distribution of sores varied by age group and

pathogen. These results highlight key areas of the body for clinicians to pay attention to when examining children for skin sores.

## INTRODUCTION

Impetigo (also known as skin sores) is a bacterial infection caused by the pathogens *Streptococcus pyogenes* (Group A Streptococcus or GAS) and *Staphylococcus aureus* (*Carapetis et al., 2005*; *World Health Organization, 2005*). The disease is estimated to affect more than 162 million people worldwide with remote Indigenous communities of Australia regarded as one of the highest areas of disease prevalence (*Carapetis et al., 2005*; *Bowen et al., 2015*; *May, Bowen & Carapetis, 2016*). An estimated 16,000 Indigenous children are affected at any one time (*Bowen et al., 2015*). Risk factors for developing impetigo are wide ranging and include tropical climate, overcrowding, poverty, scabies co-infection, insect bites and trauma (*World Health Organization, 2005*).

Skin sores can result in a number of serious complications with high rates of morbidity and mortality (*Carapetis et al., 2005*; *May, Bowen & Carapetis, 2016*; *McDonald, Currie & Carapetis, 2004*; *McDonald et al., 2008*). These include GAS and *S. aureus* septicaemia, rheumatic fever with or without rheumatic heart disease, and acute post-streptococcal glomerulonephritis (*Carapetis et al., 2005*; *May, Bowen & Carapetis, 2016*; *McDonald, Currie & Carapetis, 2004*). The adverse health effects also result in substantial socioeconomic consequences. Costly treatments impose a significant economic burden to healthcare systems while absent school days and high transmission rates reinforce the cycle of poverty and low socioeconomic status (*World Health Organization, 2005*).

Clinical disease generally presents as crusted papules that can arise in bullous or non-bullous forms (*Yeoh, Bowen & Carapetis, 2016*). GAS has been consistently shown to be the most common causative pathogen in tropical climates while more temperate settings show an increased prevalence of *S. aureus* (*Bowen et al., 2015*; *Yeoh, Bowen & Carapetis, 2016*). There is also growing concern over the increasing rates of community-acquired methicillin resistant *S. aureus* (CA-MRSA) causing skin sores (*Bowen et al., 2015*; *Tong et al., 2011*; *Bowen et al., 2014a*; *Tong et al., 2015*). Treatment options include topical antimicrobial agents for limited impetigo and systemic therapy for more extensive disease or in areas of high prevalence (*Yeoh, Bowen & Carapetis, 2016*; *Bowen et al., 2014a*).

The broad anatomical distribution of skin sores has been understood as affecting the lower limbs more commonly than the upper limbs and other body regions (scalp, face, neck and torso) (*Bowen et al., 2015*; *Bowen et al., 2014a*). However, more detailed descriptions from endemic regions are currently lacking and whether certain disease subgroups are associated with specific skin sore anatomical distribution has not yet been explored.

## MATERIALS AND METHODS

### Study design & aims

We used data from a recent randomised control trial (RCT) comparing oral co-trimoxazole with intramuscular benzathine benzylpenicillin for the treatment of impetigo in remote Indigenous communities of the Northern Territory (NT), Australia (*Bowen et al., 2014a*). To determine associations between demographic and pathogen factors with anatomical distribution of sores, we generated heat-map distributions of skin sores on the human body from 56 predefined anatomical locations and stratified skin sore distribution based on the variables sex, age, causative pathogen and scabies co-infection.

### Data & definitions

The RCT included 508 Indigenous children aged 3 months to 13 years from seven remote communities of the NT, Australia (*Bowen et al., 2014a*). Children could be re-recruited if they presented with new episodes of sores if it was greater than 90 days after a previous presentation. As such, there were 663 skin sore episodes and data from all episodes was included.

As per the RCT protocol, sores were defined as 'mild impetigo' if there was the presence of one purulent or crusted sore and less than five sores in total, and 'severe impetigo', as the presence of two or more purulent or crusted sores or five or more sores in total at each episode (participant visit). The anatomical location of the most severe sores for each episode (one if 'mild impetigo', denoted 'Site A'; two if 'severe impetigo', denoted 'Site A' and 'Site B') were photographed at days 0, 2 and 7. The day 0 skin sore locations were mapped to pre-defined anatomical regions assigned numbers 1–56 corresponding to human-body templates which were included in the trial Case Report Forms (*Bowen et al., 2014a*). Microbiological samples corresponding to sites A and/or B for all sores were collected, stored and cultured as per the trial protocol (*Bowen et al., 2014a*). Whether GAS, *S. aureus*, both, or neither pathogen were recovered by culture was recorded. Each child was also examined for the presence of scabies, and each sore was classified as to whether directly associated with scabies.

### Mapping

We mapped the recorded location of sores onto front and back human body shapefiles that were sourced from a Creative Commons Attribution Licence (*Yerramilli et al., 2017*). Coordinates approximating the central areas of the 56 anatomical regions were created within the electronic shapefiles using QGIS (*QGIS.org, 2018*). These points were then assigned attribute data corresponding to the total number of skin sores and for the number of sores stratified by each variable under investigation (sex, age, causative pathogen and scabies co-infection). Drawing tools were utilised to create borders for each anatomical region to emulate the hardcopy template maps.

### Statistical analysis

We used an inverse distance-weighted interpolation (IDW) (weighting coefficient = 2.0) with a continuous colour gradient from yellow to dark purple to illustrate a low to high

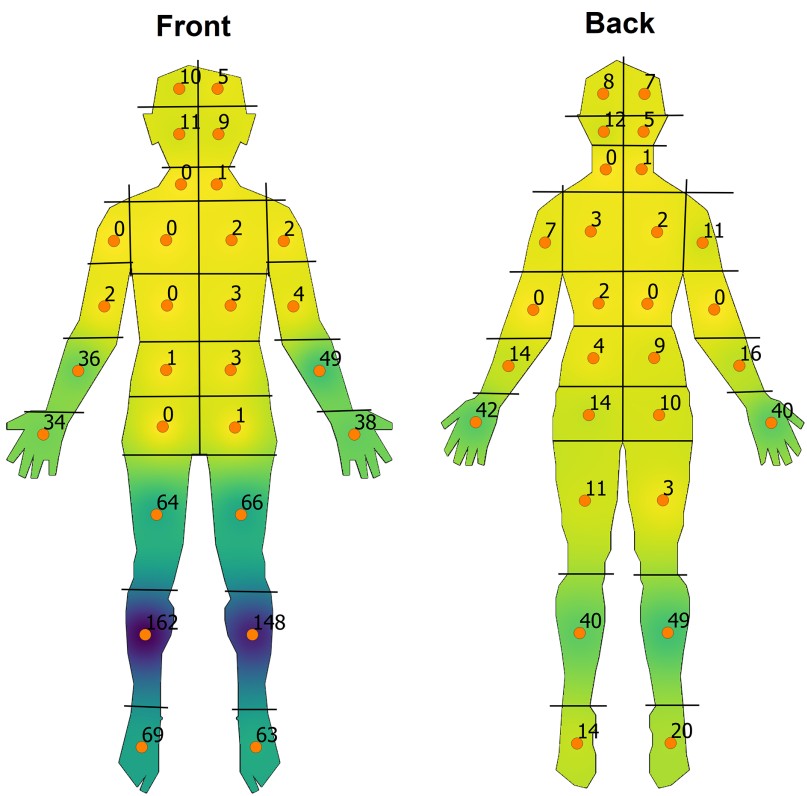

**Figure 1 Overall heat-map visualisation of the distribution of impetigo on front and back human-body shapefiles.** A continuous colour gradient from yellow to dark purple was used corresponding to areas of a low to high density of sores respectively.

density of sores respectively. To ensure standardisation among density distributions, each front and back map for each variable under consideration was assigned the same visualisation parameters including the same colour ramp, a continuous mode and a total of 52 different colour classes.

The range of pixel values ascribed to a given colour output was unique among pairs of front and back maps for each variable of interest. This starts at a lower limit of 0 with an upper limit corresponding to the maximum value of sores on either the front or back map regardless of body region. For example, for the front and back maps of the 'male' variable of interest, the range of pixel values was 0 to 80 given that the numerical value of 80 was the highest number of sores for either the front or back map, which corresponded to the 'right anterior leg' region (region number 17, Fig. 1).

This was to allow for better colour discrimination among each variable while retaining the same scale for any given front and back map pair while also still allowing for qualitative comparisons between variables. The resultant heat maps were clipped to the boundaries of the template shapefiles and each anatomical region was labelled with the number of sores with totals corresponding to overall trends or depending on the variable under consideration.
Fisher's Exact Test (FET) with a *p*-value computed by Monte Carlo simulation using 100,000 replicates was used to compare the distribution of sores between males and females; between sores with only GAS and sores with only *S. aureus*; and across all ages, with these comparisons chosen *a priori*. The number of replicates was chosen to ensure that the *p*-value estimate had stabilized to at least the third decimal place. Statistical analysis was carried out in R version 4.0.5 (*R Core Team, 2021*).

## Ethics

This was a low/negligible risk ethics endeavour using de-identified data sourced from the Skin Sore Trial. The Northern Territory Department of Health and Menzies School of Health Research Human Research Ethics Committee provided approval for the original trial (HREC 09/08) and subsequent use of the data (HREC 15-2516).

# RESULTS

## Cohort analysis

There were 508 children with 663 episodes of impetigo during 2009–2012. There was data for a total of 1,127 sores (Table S1). For these 1,127 sores, 569 were in males and 558 in females. Of the 508 children, the median age was 7.1 years (IQR = 5–9) (48% girls, 52% boys), while the range was 3 months to 13 years. Out of the total number of sores, children aged 0–2 years comprised 104 sores (9.2%), 3–6 years had 413 sores (36.6%), 7–9 years had 373 sores (33.1%), and 10–13 years comprised 237 sores (21.0%). Group A *Streptococcus* was isolated from 918 sores (81.5%), *S. aureus* from 776 sores (68.9%) while GAS and *S. aureus* together were cultured in 662 (58.7%) samples. GAS was the only isolate recovered in 256 sores (22.7%) and *S. aureus* only, in 93 sores (8.3%). Other *Streptococcus* organisms made up 1.9% of sores. A total of 103 children were classified as having a scabies infection at the time of examination while 66 (5.9%) sores were found to be directly associated with scabies lesions.

## Overall distribution

The lower limbs were the most affected body sites followed by the distal upper limbs, face and scalp (Fig. 1 & Table S1). The head, neck and torso were relatively preserved with only very small numbers of skin sores in these areas. On the anterior surface of the body, the pre-tibial region was most affected while on the posterior surface, the dorsum of the hands and calves predominated.

## Stratification by variables

Stratification by sex revealed a similar overall distribution pattern of sores with no observable difference between males and females (Fig. S1; FET, *p* = 0.154, degrees of freedom (df = 55)). Young children and infants aged 0–2 years had more evenly distributed densities on the posterior surface of the body compared to other age groups (Fig. S2). Children aged 3–6 years start to show a more typical distribution pattern but still have relatively high densities on the upper posterior lower limbs and scalp regions compared to older age groups. Those in the 7–9 and 10–13 years of age groups display a distribution

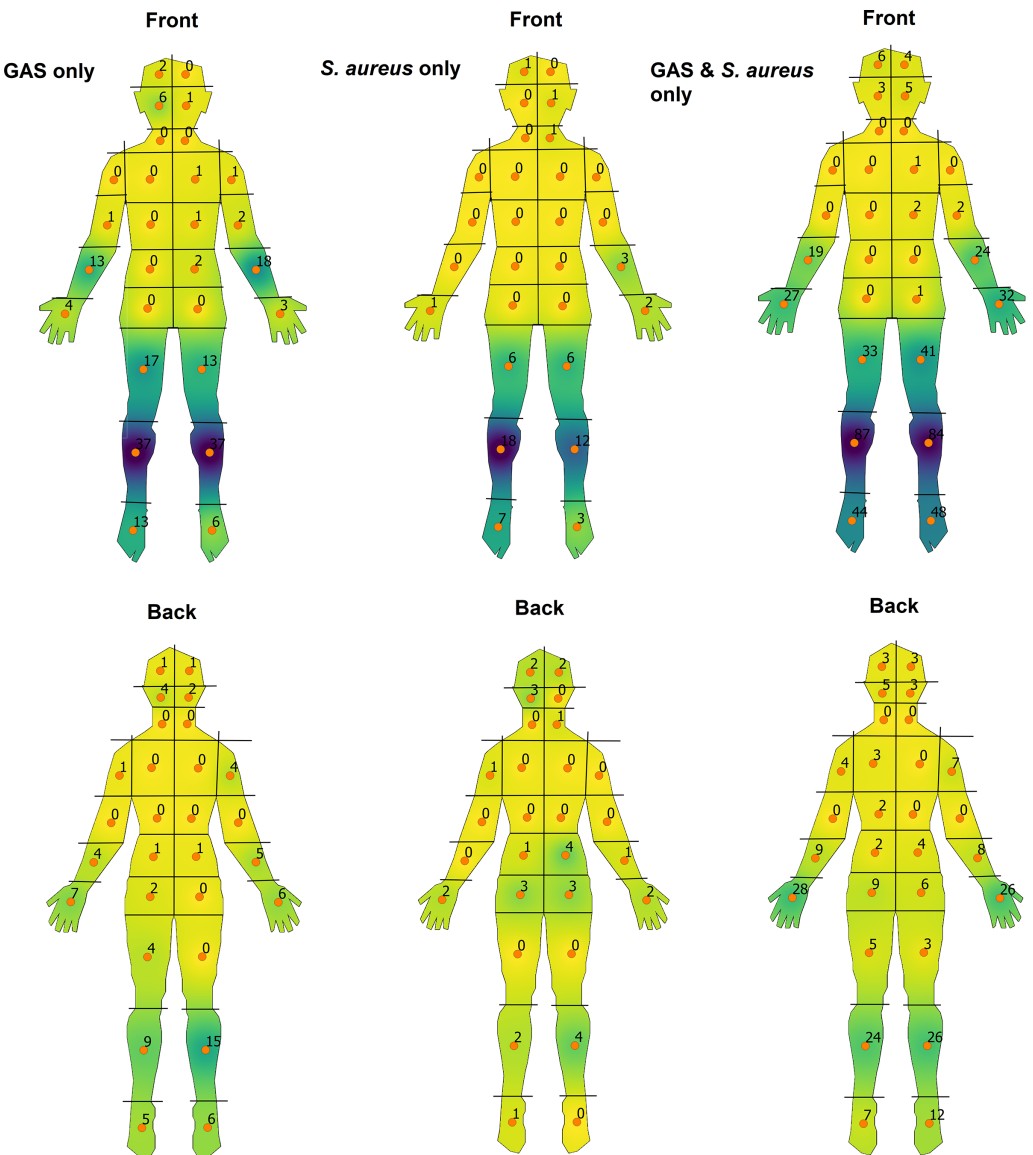

**Figure 2 Heat-map visualisation of the body-distribution of impetigo stratified by causative pathogen.** A continuous colour gradient from yellow to dark purple was used corresponding to areas of a low to high density of sores respectively.

pattern similar to that for overall sores as described above. Overall, differences in the patterns of sore distribution by age were statistically significant (FET, $p = 3 \times 10^{-5}$, df = 165).

Swabbed sores which cultured both GAS and *S. aureus* did not show any major observable differences from overall distribution patterns as did those samples which grew GAS only (Fig. 2). Isolates of *S. aureus* only were found to be more common on the upper posterior lower limbs when comparing to GAS-only sores, with statistically significant differences in the distribution of sores between these groups (FET, $p = 3 \times 10^{-4}$, df = 55). Skin sores determined to be directly associated with scabies infection predominantly

affected the hands (*n* = 66; Fig. S3), while participants who were noted to have concurrent scabies, regardless of whether associated with their skin sores, showed predominance for the lower limbs (*n* = 192; Fig. S4). Head lice co-infection predominantly affected the scalp, while those with tinea co-infection showed sporadic distribution although the sample size for each of these variables was low (*n* = 10 and *n* = 6 respectively).

## DISCUSSION

We have provided a detailed description of the anatomical locations of impetigo lesions in children from an area with endemic rates of impetigo. Exposed areas of the lower and upper limbs were the most commonly affected areas, followed by relatively equal distribution on the face, scalp and upper posterior lower limbs. We found no observable difference between males and females and illustrate that younger children have more evenly dispersed sores on the posterior surface of the body, especially at the lower limbs and scalp.

Overwhelmingly, the pattern of distribution appears to affect more exposed body surfaces, such as the distal lower and upper limbs, as well as the face and scalp when compared to the torso specifically. The increased propensity of impetigo in these areas corresponds with the occurrence of bacterial superinfection after insect bites and trauma (*World Health Organization, 2005*). Regarding the limbs, these observations would seemingly fit with observed behaviour of children often with uncovered upper and lower limbs due to hot climate. Many of these children then engage in bare-footed play which may result in minor trauma and skin abrasions. Further, scabies co-infection was almost exclusively limited to the hands and corresponds to previously described patterns of scabies infection with resultant pyoderma (*Yeoh, Bowen & Carapetis, 2016*). This may suggest impetigo of the hands could be largely due to superinfection of burrows of the scabies mite as opposed to scabies-induced pruritus leading to breaks in skin integrity and resultant pyoderma at other body sites.

The call for more targeted preventative measures for impetigo may be further supported by our results. While treatment efforts in high prevalence areas have had substantial impact, encouraged by methods to reduce the cycle of poverty and socioeconomic disadvantage such as improved hygiene practices and appropriate housing, additional strategies are needed to further lower the significant burden of disease that remains (*Carapetis et al., 2005*; *Bowen et al., 2015*; *May, Bowen & Carapetis, 2016*). One method is earlier detection and treatment, and standardised programs such as community skin days and mass drug administration have shown promise (*May, Bowen & Carapetis, 2016*). Our findings could augment these programs by reminding clinicians to pay particular attention to the body sites most affected, including consideration of a patient's age group. Furthermore, the use of more protective clothing such as longer upper and lower limb garments as well as encouraging the use of footwear may help reduce skin trauma thereby reducing infection. Impetigo may spread among an individual or *via* household contacts through cross contamination, and it is not uncommon that those affected have multiple sores (*Avire, Whiley & Ross, 2021*; *Creech, Al-Zubeidi & Fritz, 2015*). In this regard, earlier treatment with topical antimicrobial agents could be considered, for example, at body sites

with known increased pyoderma distribution which may have a suspicious mild lesion or break in skin integrity after diagnosis at an initial, single body site (*Koning et al., 2012*; *George & Rubin, 2003*). However, broader use of topical antimicrobial agents has been associated with the spread of antimicrobial resistance among populations with a high prevalence of skin sores (*Williamson et al., 2014*). Appropriate surveillance of antimicrobial resistance should be implemented if widespread topical antimicrobials were to be recommended.

Appropriate examination of children for impetigo is important for both opportunistic clinical encounters and for healthy skin screening programs. While exposed body sites in the lower and upper limbs are easily accessible and where most lesions occur, our findings that the buttock and scalp regions are also commonly involved highlight the importance of examining these sites, particularly in younger children. The body maps in our study may be useful for surveillance programs. Changes at the population level in body site distributions of impetigo may indicate shifts in underlying risk factors. For example, an increasing distribution on the hands may indicate an increased prevalence of scabies co-infection. Likewise, an increasing proportion of skin sores distributed in the upper posterior lower limbs may indicate more *S. aureus* related infections, especially in younger children with nappy-related dermatitis (*Ferrazzini et al., 2003*; *Heath, Desai & Silverberg, 2009*). This is in contrast to GAS-only sores which show a distribution pattern similar to overall trends. One reason for this could be that the proportion of GAS-only sores was nearly three times greater than that of *S. aureus*-only sores and therefore may better reflect the general anatomical distribution. While many risk factors between GAS and *S. aureus* skin disease overlap such as close household contact, poor hygiene and compromised skin integrity, others such as age of acquisition, seasonal trends and varying geographical prevalence could also account for the minor differences in anatomical distribution we have observed and warrants further exploration (*Bowen et al., 2015*; *Avire, Whiley & Ross, 2021*; *Bowen et al., 2014b*).

Our study has some limitations. The data we have used is limited by the setting, definitions and exclusion criteria of the Skin Sore Trial (*Bowen et al., 2014a*). This includes limiting the analysis in our study to the two most severe sores if there were multiple sores noted for a participant. While multiple sores (greater or equal to 2) was commonly noted within the trial and may impact our results, our aim was to illustrate the overall trends of impetigo body distribution and allow for a standardised comparison between key variables. Further, compared to older children within the trial, infants may have been more likely to have external clothing removed during examinations, hence resulting in a bias to finding skin sores in the upper posterior lower limbs. Our results may also not accurately reflect skin sore body distribution in other regions of Australia and other countries given data was collected from remote Aboriginal communities of the Northern Territory. The distribution of skin sores in temperate and high-income countries is likely to be different (*Bowen et al., 2015*). Where impetigo is principally caused by *S. aureus*, the distribution is also likely to be different. Ectoparasite infections can be difficult to diagnose and both scabies and head lice may have been under-diagnosed. However, the diagnoses were made in the context of a

prospective clinical trial with trained staff and the rates of scabies were in keeping with other studies from a similar setting (*La Vincente et al., 2009*; *Andrews et al., 2009*).

## CONCLUSIONS

We have visually represented the distribution of impetigo on the human body in remote Indigenous communities of the Northern Territory, Australia, with stratification by sex, age and causative pathogen. These results highlight key areas of the body for clinicians to pay attention to when examining children for impetigo.

## ACKNOWLEDGEMENTS

We thank the participants and families who contributed to the Skin Sore Trial.
We acknowledge our partners in this work: Northern Territory Remote Health, Aboriginal Medical Services Alliance Northern Territory, Northern Territory Centre for Disease Control, One Disease, Miwatj Health and the NHMRC-funded HOT NORTH initiative. We acknowledge the Lowitja Institute and the Cooperative Research Centre for Aboriginal Health who originally funded and lent significant support to the East Arnhem Healthy Skin Project. We also thank Bart Currie, Ross Andrews, and Malcolm McDonald as key investigators for the original Skin Sore Trial.

### Funding

The authors received no funding for this work.

### Competing Interests

Steven YC Tong is an Academic Editor for PeerJ.

### Author Contributions

- Arvind Yerramilli conceived and designed the experiments, performed the experiments, analyzed the data, prepared figures and/or tables, authored or reviewed drafts of the article, and approved the final draft.
- Asha C. Bowen conceived and designed the experiments, analyzed the data, authored or reviewed drafts of the article, and approved the final draft.
- Adrian J. Marcato conceived and designed the experiments, authored or reviewed drafts of the article, and approved the final draft.
- Jodie McVernon analyzed the data, authored or reviewed drafts of the article, and approved the final draft.
- Jonathan R. Carapetis analyzed the data, authored or reviewed drafts of the article, and approved the final draft.
- Patricia T. Campbell conceived and designed the experiments, performed the experiments, analyzed the data, prepared figures and/or tables, authored or reviewed drafts of the article, and approved the final draft.

- Steven YC Tong conceived and designed the experiments, performed the experiments, analyzed the data, prepared figures and/or tables, authored or reviewed drafts of the article, and approved the final draft.

## Human Ethics

The following information was supplied relating to ethical approvals (*i.e.*, approving body and any reference numbers):

This was a low/negligible risk ethics endeavour using de-identified data sourced from the Skin Sore Trial. The Northern Territory Department of Health and Menzies School of Health Research Human Research Ethics Committee provided approval for the original trial (HREC 09/08) and subsequent use of the data (HREC 15-2516).

## Data Availability

The raw data used for the heatmaps is available in the Supplemental File.

## Supplemental Information

Supplemental information for this article can be found online at http://dx.doi.org/10.7717/peerj.14154#supplemental-information.

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
