# Peer review of "Body distribution of impetigo and association with host and pathogen factors"

_PeerJ, doi:10.7717/peerj.14154_

## Round 0.1 · original submission · Minor Revisions

Three Reviewers assessed your manuscript and are positive about its publication in this journal. One has no comments to address, while the other two have minor points.

Reviewer 1 ·

Basic reporting

The paper is well laid out with clear description of the overall methodolgy

Experimental design

No issues. My only thought is whether formal statistical analysis is really adding much here compared to the very nice chlorapleth maps? I dont feel strongly but I personally found the visual display of information more useful than the fishers-exact results!

Validity of the findings

As noted there are some limitation due to the underlying population the study data is drawn from but this is well described.

If a patient had impetigo in 3 locations in the original trial am I correct that only 2 would have been recorded? Do you have any sense of how common this would be? It seems a potential limitation.

Additional comments

This is really a very nice article and my comments are minor. It could probably be accepted as is or after minor changes only.

Reviewer 2 ·

Basic reporting

No comment

Experimental design

No comment

Validity of the findings

No comment

Additional comments

A well written article, covering an important clinical condition. I commend the authors for pursuing an area of research that is under-explored.

Reviewer 3 ·

Basic reporting

Overall, the research is a good addition to the literature given it brought a novel approach to demonstrate body distribution of impetigo. But the work and its overall relevance can be substantiated using more references etc.

Methods
• Given ~60% of children had co-infection by S. Pyogenes and S. aureus, I wonder if there was any consideration of this on the analyses? Because it appears based on your results the individual pathogens are identified only in 30% of the overall sores in the analysis, leaving the majority of sores not considered for the mapping?

Results
• S1 fig is stratification by age group but reported for gender on line 179 of the manuscript.
• Also, double check S2 Fig with what you have referred in the manuscript.
• Overall, check all your references to the figures. Seems like most of them are not referring to the right figures or appendixes.
• Line 161 put a full stop after 13 years.

Discussion
• The discussion only has 4 or 5 new references – I feel like this can be substantiated further citing a few more works in this area.
• The body distribution with respect to the implicated pathogen, particularly the different pattern of GAS only sores, is barely discussed. Does this difference have any clinical relevance? Or any reason why this difference could exist?
• Headlice should be modified as head lice throughout.
• What is the overall implication of the varied distributions of impetigo for treatments and treatment outcomes?
• Given head lice was identified as a co-infection only in 10 cases, I wouldn’t say the scalp distribution is common.

Experimental design

No comment

Validity of the findings

Overall, the findings are usefull for clinicians and researchers to understand the disease distribution and its implication.

Additional comments

None.

---

## Round 0.2 · accepted · Accept

The authors addressed all the Reviewers' concerns.

---

## Author Rebuttal · Round 0.2

Dr Arvind Yerramilli

Department of Infectious Diseases

Barwon Health

Ryrie Street, Geelong

VIC 3220, Australia

PeerJ, Inc.

PO Box 910224

San Diego

CA 92191

USA

PeerJ, Ltd.

727-729 High Road

London, N12 0BP

UK

Dear Academic Editor, Professor Hector Mora-Montes and the Reviewers,

Re: **'Body distribution of impetigo and association with host and pathogen factors'**

Thank you for your time reviewing our manuscript. We appreciate the feedback and comments on how we can further strengthen our submission.

Please find below a detailed list of responses and description of changes in the manuscript. Please note that all references to line numbers in our responses refer to the 'tracked changes' version of the manuscript.

Thank you for taking the time to read and consider this resubmission. We look forward to hearing from you.

Yours sincerely,

Dr Arvind Yerramilli
* * *
Reviewer 1

*Basic reporting*

*The paper is well laid out with clear description of the overall methodolgy*

*Experimental design*

*No issues. My only thought is whether formal statistical analysis is really adding much here compared to the very nice chlorapleth maps? I dont feel strongly but I personally found the visual display of information more useful than the fishers-exact results!*

Thank you for the kind feedback. We agree the chlorapleth maps are the main highlights of the study and allow for qualitative comparison between variables however we felt the statistical analysis added some additional quantitative backup to the visual assessment between key variables.

*Validity of the findings*
*As noted there are some limitation due to the underlying population the study data is drawn from but this is well described.*

*If a patient had impetigo in 3 locations in the original trial am I correct that only 2 would have been recorded? Do you have any sense of how common this would be? It seems a potential limitation.*

We did only collect anatomical location and variable data from the 2 most prominent/severe sores and agree this could be a potential limitation as children often present with multiple sores. We have now provided additional text as follows making note of this limitation in lines 262-267:

"This includes limiting the analysis in our study to the 2 most severe sores if there were multiple sores noted for a participant. While multiple sores (greater or equal to 2) was commonly noted within the trial and may impact our results, our aim was to illustrate the overall trends of impetigo body distribution and allow for a standardised comparison between key variables."

Additional comments

*This is really a very nice article and my comments are minor. It could probably be accepted as is or after minor changes only.*

Thank you.

Reviewer 2 (Anonymous)

*Basic reporting*
*No comment*

*Experimental design*
*No comment*

*Validity of the findings*
*No comment*

*Additional comments*
*A well written article, covering an important clinical condition. I commend the authors for pursuing an area of research that is under-explored.*

Thank you for the positive feedback.

*Reviewer 3 (Anonymous)*

*Basic reporting*

*Overall, the research is a good addition to the literature given it brought a novel approach to demonstrate body distribution of impetigo. But the work and its overall relevance can be substantiated using more references etc.*

*Methods*
*• Given ~60% of children had co-infection by S. Pyogenes and S. aureus, I wonder if there was any consideration of this on the analyses? Because it appears based on your results the individual pathogens are identified only in 30% of the overall sores in the analysis, leaving the majority of sores not considered for the mapping?*

Thank you for pointing this out. On review of Figure 3 (now Figure 2, see below comment) we have noted an error with the front map of the GAS & *S. aureus* combined variable. The labels were incorrectly showing the GAS-only variable (although the heat-map output was correct). This has now been fixed to show the GAS & *S. aureus* combined labels which makes up the ~60% of sores that was noted in our results.

While we have not provided a statistical analysis comparing both *S. aureus* and *S. pyogenes* with either pathogen individually, we do map the results for *S. pyogenes* alone, *S. aureus* alone and *S. pyogenes* and *S. aureus* together (new Figure 2). We have discussed at lines 191-193 that there were no observable differences in visual comparison between sores that cultured *S. pyogenes* only and those that cultured both *S. pyogenes* and *S. aureus* and feel this will be sufficient for the reader without any extra quantitative results.

*Results*
*• S1 fig is stratification by age group but reported for gender on line 179 of the manuscript.*
*• Also, double check S2 Fig with what you have referred in the manuscript.*
*• Overall, check all your references to the figures. Seems like most of them are not referring to the right figures or appendixes.*
*• Line 161 put a full stop after 13 years.*

Thank you for pointing out these discrepancies.

Regarding figures, we have re-submitted the images and corrected references to them made in the text which should be as follows. Please note, after advice from the PeerJ team due to inability to verify certain copyright licensing, we have replaced the previous Figure 1 (hard-copy template maps) with an in-text citation and have updated the figure numbers as follows:

Figure 1 = overall heatmap distribution (Line 175)

Figure 2 = density maps stratified by pathogen (Line 192)
S1 Figure = density maps stratified by sex (Line 182)
S2 Figure = density maps stratified by age (Line 184)
S3 Figure = density maps stratified by scabies co-infection (Line 196)
S4 Figure = density maps stratified by scabies association (Line 198)

The grammatical error of the full stop on line 164 has now been corrected.

*Discussion*
*• The discussion only has 4 or 5 new references – I feel like this can be substantiated further citing a few more works in this area.*
*• The body distribution with respect to the implicated pathogen, particularly the different pattern of GAS only sores, is barely discussed. Does this difference have any clinical relevance? Or any reason why this difference could exist?*
*• Headlice should be modified as head lice throughout.*
*• What is the overall implication of the varied distributions of impetigo for treatments and treatment outcomes?*
*• Given head lice was identified as a co-infection only in 10 cases, I wouldn't say the scalp distribution is common.*

We agree the discussion could be substantiated further with more references.

As per lines 231 - 240 we have added the following text and references regarding prevention and treatments:

"Furthermore, the use of more protective clothing such as longer upper and lower limb garments as well as encouraging the use of footwear may help reduce skin trauma thereby reducing infection. Impetigo may spread among an individual or via household contacts through cross contamination, and it is not uncommon that those affected have multiple sores[14,15]. In this regard, earlier treatment with topical antimicrobial agents could be considered, for example, at body sites with known increased pyoderma distribution which may have a suspicious mild lesion or break in skin integrity after diagnosis at an initial, single body site[16,17]. However, broader use of topical antimicrobial agents has been associated with the spread of antimicrobial resistance among populations with a high prevalence of skin sores[18]. Appropriate surveillance of antimicrobial resistance should be implemented if widespread topical antimicrobials were to be recommended."

As per lines 251 – 259, we have added additional discussion regarding our findings for GAS-only sores:

"This is in contrast to GAS-only sores which show a distribution pattern similar to overall trends. One reason for this could be that the proportion of GAS-only sores was nearly three times greater than that of *S. aureus*-only sores and therefore may better reflect the general anatomical distribution. While many risk factors between GAS and *S. aureus* skin disease overlap such as close household contact, poor hygiene and compromised skin integrity, others such as age of acquisition, seasonal trends and varying geographical prevalence could also account for the minor differences in anatomical distribution we have observed and warrants further exploration[3,14,21]."

Headlice has been modified to head lice at lines 36, 199 and 274 as suggested. We have also qualified lines 212-213 to refer to the face and scalp being more affected in comparison to the torso specifically where there were hardly any sores observed.

*Experimental design*
*No comment*

*Validity of the findings*
*Overall, the findings are usefull for clinicians and researchers to understand the disease distribution and its implication.*

*Additional comments*
*None.*